# Multi-Label Active Learning-Based Machine Learning Model for Heart Disease Prediction

**DOI:** 10.3390/s22031184

**Published:** 2022-02-04

**Authors:** Ibrahim M. El-Hasnony, Omar M. Elzeki, Ali Alshehri, Hanaa Salem

**Affiliations:** 1Faculty of Computers and Information Sciences, Mansoura University, Mansoura 35516, Egypt; Ibrahimhesin2005@mans.edu.eg; 2Faculty of Computer Science, New Mansoura University, Gamasa 35712, Egypt; 3Department of Computer Science, University of Tabuk, Tabuk 71491, Saudi Arabia; a.alshehri@ut.edu.sa; 4Faculty of Engineering, Delta University for Science and Technology, Gamasa 35712, Egypt; Hana.salem@deltauniv.edu.eg

**Keywords:** heart disease, active learning, multi-label classification, chronic diseases, data mining, machine learning

## Abstract

The rapid growth and adaptation of medical information to identify significant health trends and help with timely preventive care have been recent hallmarks of the modern healthcare data system. Heart disease is the deadliest condition in the developed world. Cardiovascular disease and its complications, including dementia, can be averted with early detection. Further research in this area is needed to prevent strokes and heart attacks. An optimal machine learning model can help achieve this goal with a wealth of healthcare data on heart disease. Heart disease can be predicted and diagnosed using machine-learning-based systems. Active learning (AL) methods improve classification quality by incorporating user–expert feedback with sparsely labelled data. In this paper, five (MMC, Random, Adaptive, QUIRE, and AUDI) selection strategies for multi-label active learning were applied and used for reducing labelling costs by iteratively selecting the most relevant data to query their labels. The selection methods with a label ranking classifier have hyperparameters optimized by a grid search to implement predictive modelling in each scenario for the heart disease dataset. Experimental evaluation includes accuracy and F-score with/without hyperparameter optimization. Results show that the generalization of the learning model beyond the existing data for the optimized label ranking model uses the selection method versus others due to accuracy. However, the selection method was highlighted in regards to the F-score using optimized settings.

## 1. Introduction 

Hospitals and clinics are constrained to storing and analyzing medical data using traditional and manual methods. Many medical institutions have made significant efforts to overcome this limitation by combining considerable data resources with new technologies [1]; however, numerous medical facilities failed to implement new systems early. There is still a lack of knowledge about diseases and how to treat them, despite the enormous number of data available. In light of the data’s complexity, data mining and machine learning (ML) techniques [2] are becoming increasingly popular for their use in data analysis. Machine learning and data-driven tactics can produce accurate diagnostic tools. The current study aims to identify and assess the organizational hurdles that prohibit medical institutions from adopting a successful method to offer managers strategic solutions to these difficulties [3]. 

Active learning approaches are increasingly becoming new and fascinating instruments for evaluating healthcare data due to their success in numerous sectors and their rapid and continual methodological improvement. Studies show that unlabelled data are more common than labelled data in the actual world. Because label acquisition is often expensive due to the involvement of human specialists, it is vital to train an accurate prediction model with a small number of labelled cases. AL selects only the most valuable examples for class assignments to overcome this problem. Active learning is one of the most common methods for gaining knowledge from sparsely labelled data [4]. It aims to cut down on the time needed to annotate data by searching for the most relevant examples. Unlabelled data abounds, yet labelling is too expensive. Choosing acceptable criteria for determining which instances are worth querying is crucial when designing an active learning algorithm. Active learning algorithms commonly use informativeness and representativeness as two query selection criteria. An instance’s informativeness measures its ability to reduce statistical model uncertainty. In contrast, an instance’s representativeness measures its ability to accurately represent unlabelled data input patterns [5].

One of the most critical and challenging issues in modern medicine is accurately predicting the onset of heart disease. Heart disease is the leading cause of death in many developed nations. Approximately one in every four people who pass away each year in the United States is a victim of this. Cardiovascular disease weakens the body, especially in adults and the elderly, because it affects blood vessel function and results in coronary artery infections [6,7,8]. 

Many algorithms for predictive learning are available (e.g., linear and logistic regression, classification and regression trees, learning vector quantization (LVQ), support vector machines (SVM), boosting, and deep neural networks). Their overarching goals and constraints differ. ML-based analyses frequently look for nonlinear relationships among tens or hundreds of thousands of different variables. For these methods to be most effective, a massive proportion of data for training is necessary. When the information is plentiful, but the labels are complicated, time-consuming, or expensive to obtain, the use of active learning would be highly useful. 

Hybrid models based on data mining techniques are currently being used to predict and diagnose cardiovascular disease [9,10]. The hybrid model combines two methods that work better than any single method. A logistic-regression-based prediction model has been utilized to diagnose cardiac disease [11]. Medical imaging has also benefited from the use of ML in the discovery of object features automatically [12]. 

In this paper, the proposed model tries to solve the problem of memorizing learning models. Active learning is a perfect method if the model does not overfit the data instances. However, it is still good to train only on samples that significantly impact its performance. Hence, the goal was to achieve a model that generalizes the existing data, i.e., not memorization, but a generalization. In many cases, especially in high-dimensional settings, learning a model that works well on training data but fails on new data is common. 

In many cases of regular machine learning algorithms, the model has “overfit” or “underfit” the training data (i.e., it has simply memorized/unmemorized the data). So, five selection strategies for multi-label active learning are applied. The five methods are MMC, Random, Adaptive, QUIRE, and AUDI. The grid search with the label ranking classifier is implemented as the predictive modelling in each strategy. There have been several different conditions under which the system can be stopped. As a rule of thumb, the AL procedure is repeated several times (number of iterations). The base classifier’s performance is evaluated using a test set and an evaluation metric. The entire model was applied to the dataset of heart disease. As a result of the research, it was concluded that the learning model could be extrapolated to include new data.

In turn, the proposed manuscript seeks objectives that include illustrating the effectiveness of the active learning algorithms for diagnosing heart disease. Active learning generalizes the concepts of the outcomes and findings to be interoperated for any new case(s) without the need to be a valid instance of the training, test, or validation samples. A second objective includes the hyperparameters’ optimization for the active learning algorithms using the grid search method. In turn, the key significant contributions of the proposed work include: The empirical statistical analysis of heart disease using different visualization charts;The primacy of applying active learning for diagnosing heart disease also enhances generalization over the memorization of the generated model;Comparative evaluation of five active learning selection strategic methods versus the regular machine learning algorithms;Optimizing the hyperparameters of the trained model using the grid search method.

Here is how the rest of the paper is structured: Section 2 summarizes recent related works, Section 3 outlines our proposed approach methods, Section 4 introduces experimental findings, and Section 5 wraps up the paper by discussing possible future directions.

## 2. Related Work

### 2.1. Heart Disease 

Healthcare has been a significant field of research for the last decade. Almost all algorithms are implemented and tested positively in the healthcare domain [13]. Though medical cardiology is critical, recent advancements in data mining and machine learning techniques have created significant diverse domains [14]. Many medical data are accumulated every day, and researchers have tested their algorithms [15]. Developing countries suffer from a significant number of deaths caused by heart malfunctioning [16,17]. There are many studies on manipulating heart disease diagnosis and prediction. A new end-to-end deep learning method for diagnosing heart diseases from a single-channel ECG signal was presented by Khalil et al. [18]. Patients’ heart sounds can be monitored in real-time, and any abnormalities can be detected, thanks to a digital stethoscope prototype developed by Chowdhury et al. [19]. Heart disease classification was improved using the fast-correlation-based feature selection (FCBF) method developed by Khourdifi et al. [20]. This was followed by other classification methods, including Naïve Bayes, support vector machine (SVM), K -nearest neighbour (KNN), random forest (RF), and a multilayer perception artificial neural network optimized using ant colony optimization techniques (ACO) and particle swarm optimization (PSO). Latha et al. combined multiple heart disease classifiers in their research [21]. Similarly, by merging multiple classifiers, we enhanced the accuracy of weak algorithms. Li et al. [22] suggested an ML-based approach for identifying heart disease. Decision trees, neural networks, K-nearest neighbours, and support vector machines are all examples of classification methods.

In contrast, common feature selection techniques such as Relief, minimal redundancy maximal relevance, the minor absolute shrinkage selection operator, and local learning for reducing redundant and irrelevant features were employed for the system’s development. To solve the problem of feature selection, they proposed a novel fast conditional mutual information algorithm. 

In this paper, the heart disease dataset was evaluated with active learning methods [23]. Many machine-learning-based studies have used this dataset for heart disease prediction and classification. In these studies, models were based on logistic regression; for example, in [24], Khanna et al. conducted a comparative assessment of widely used machine learning algorithms to predict heart disease prevalence. The classification methods are based on the Cleveland Dataset, which is freely available. Different models ertr compared, and their ability to predict cardiac disease was evaluated. Khan et al. used ZeroR [25] to examine numerous machine learning algorithms using a heart disease dataset (diameter narrowing) to predict angiographic disease status. Achayra et al. [26] used multiple criteria to compare the algorithms that can appropriately classify heart disease. Sarangam et al. [27] recommended a heart disease prediction system (HDPS) based entirely on data mining techniques using the Naïve Bayes algorithm. Kumar et al. [28] employed K-star, J48, SMO, Naïve Bayes, MLP, random forest, Bayes net, and REPTREE data extraction strategies to predict heart problems. Tougui et al. [29] used six widely used data mining tools: Orange (logistic regression), Weka (support vector machine), RapidMiner (k-nearest neighbours), KNIME (Matlab), and scikit-Learn (random forest) and compared these tools with six widely used machine learning techniques to classify heart disease. However, recent proposals have been introduced that specifically aim to learn the relation of the features and labels of the dataset [23]. These are memorization-based methods and algorithms based on the current knowledge of the authors; active learning to handle the generalization concept using a revelent medical dataset has not yet been introduced. Hence, we introduce a model that aims to demonstrate the efficacy of active learning algorithms in diagnosing heart disease. There is no requirement that any new case must be a valid instance of the training, test, or validation samples when using active learning to generalize the concepts of the results and findings.

### 2.2. Active Learning 

Two key ideas drive active learning research: (i) Learners should be allowed to ask questions, and (ii) unlabelled data are frequently readily available or easy to obtain. Active learning aims to reduce the amount of time and money spent on labelling to develop a prediction model that can make accurate predictions. Active learners select the most informative examples from a large pool of unlabelled instances and then query an oracle (e.g., human annotator) for labels iteratively. Single-label classification is the most common active learning problem studied in the literature. Uncertainty sampling, in which the learner labels the most ambiguous instance for a previously trained classification model, is a standard active learning strategy. Uncertainty sampling methods are computationally efficient. They have shown good empirical performance, even though they do not measure the future predictive informativeness of the candidate instance on the large amounts of unlabelled data [30]. Using the evolutionary algorithm USPEX and machine-learning interatomic potentials actively learning on the fly, Podryabinkin et al. [31] proposed a method for crystal structure prediction. As implemented in the MLIP (machine-learning interatomic potentials) package, active learning was used by Novikov et al. [32] to construct moment tensor potentials, with a focus on the most efficient ways to automatically sample configurations for the training set. 

### 2.3. Active Learning in Healthcare

Automated hyperparameter selection was presented by Owoyele et al. [33] in conjunction with an active learning approach. They used a Bayesian approach to optimize the hyperparameters of the base learners that make up a super learner model. Using simulations, machine learning training, and surrogate optimization, they used an active learning approach to refine the solution near the predicted optimum. Automated dataset generation for training universal machine learning potentials for molecular energetics was presented by Smith et al. [34]. Inferring the accuracy of an ensemble’s prediction is based on the concept of active learning (AL), which is implemented through query by committee (QBC). A new data-driven approach to AL was proposed by Konyushkova et al. [35]. Regressors can be trained to predict how much error reduction a candidate sample can expect to see in a specific learning state. Instead of learning from previous AL results, they can use strategies based on previous AL outcomes because they formulated the query selection procedure as a regression problem. Recent advances in Bayesian deep learning have been incorporated into an active learning framework by Gal et al. [36]. The researchers developed an active learning framework for high-dimensional data.

Furthermore, active learning has improved healthcare applications even further. Active-learning-based cross-population train–test models were developed by Santosh et al. [37] using multitudinal and multimodal data for COVID-19 detection. In a pediatric cardiac MRI for congenital heart disease, Pace et al. [38] presented an interactive algorithm for segmenting the heart chambers and epicardial surfaces, including the great vessel walls. When segmentation error was likely, they looked into using active learning to solicit user input automatically. Ghosh et al. [39] used the Cleveland Heart Disease dataset to develop an intelligent diagnostic framework for predicting heart disease. A variety of feature sets were combined with three machine learning approaches: decision tree (DT), K-nearest neighbour (KNN), and random forest (RF). All features were subjected to “Pearson’s Correlation” and the Relief algorithm, which selected ten features from the larger pool. The proposed work relies heavily on active learning models for data classification, which are highly effective and popular.

## 3. Research Methods

In this section, a detailed description of the implemented methods and techniques is given. Firstly, we introduce machine learning and the challenges related to supervised learning. Secondly, we provide a detailed description of active learning and the selected methods. Finally, we present the step-by-step procedure for the implemented model and the evaluation metrics.

## 3.1. Machine Learning 

The study of tools and methods for identifying patterns in data is called “machine learning.” Using these patterns, it is possible to learn more about the world we live in today and predict how the world will change over time, such as by identifying risk factors for infection. When it comes to machine learning, there are a few general categories: supervised, unsupervised, and reinforcment learning [40].

Supervised learning is the subject of this article because the data is “labelled” according to the desired outcome (e.g., patients are either infected or not infected). The algorithm then discovers a relationship between a set of covariates (such as patient demographics) and the outcome. The training data is used in this step. This mapping can identify or predict new test data once learned. Multi-label classification [41] guides learning a procedure that performs mapping instances x∈X to label subsets $P_{x}\subset ℒ$, where $ℒ=\left\{\lambda_{1},\ldots ,\lambda_{c}\right\}$ is a delimited set of predefined labels, typically with a small to a moderate number of alternatives. Thus, in multiclass learning, possibilities are not assumed to be mutually exclusive. Multiple labels may be associated with a single instance. The collection of labels $P_{x}$ is relevant for the conveyed instance; the set $\;N_{x}=ℒ\setminus P_{x}$ represents unrelated labels.

For a learning algorithm, selecting the optimal hyperparameters is known as hyperparameter tuning in machine learning. It is possible to manipulate the learning process using hyperparameters. On the other hand, the values of other parameters (usually node weights) are predetermined in advance. Hyperparameter optimization has traditionally been performed using a grid search, or a parameter sweep, an exhaustive search of a manually chosen subset of the hyperparameter space. A grid search algorithm must be directed by some performance metric, such as cross-validation on the training set or evaluation on a holdout validation set. Some parameters in a machine learning system may have real-valued or unbounded-value spaces that need to be discretized before a grid search can be applied.

Numerous learning algorithms exist to complete the task (e.g., logistic regression, decision trees, ensemble approaches, and deep neural networks). Objective function and constraints are the most apparent differences between these approaches. Despite their close ties to traditional statistics, ML-based analyses often seek nonlinear relationships among hundreds or thousands of covariates. As expected, these methods work best when they have a large number of “training” data (i.e., when there are many examples to learn from). In this case, the goal is to learn a model that can be applied to a wide range of situations to generalize, not to memorize. To rank instances according to a set of specified labels, one must perform a complicated prediction task known as label ranking. Multiclass prediction, multi-label classification, and hierarchical classification are subsumed by this topic, which is fascinating. There are many cases where learning a model has led good results on the training data but fails when applied to never-before-seen data, especially in settings where there are hundreds or thousands of covariates (i.e., high-dimensional settings). Such a model is said to be “overfitting” the data it was trained on (i.e., it has simply memorized the data). There are various ways to deal with these issues, and they all depend on the learning framework.

## 3.2. Active Learning Modelling 

A multi-label issue has a feature space $F_{space}$ and a label space ${ℒ}_{space}$, both of which have cardinality n. (label number). We adopt the pair $X_{i},Y_{i}$ to represent a multi-label instance of an example i where $X_{i}$ is the feature vector and $Y_{i}$ is the binary vector. Assume $Y_{i}$ is a binary vector with n components, with $Y_{i\ell }\;$ indicating whether the example i belongs to the n-th label.

We can say that ȹ is a multi-label classification algorithm that simultaneously handles both multi-label classification and label ranking tasks. So for a specified test example, (i) ȹ decomposes the label space ${ℒ}_{s}$ into related and unrelated labels (positive label(s)) and (ii) ȹ produces label ranking depending on their relevance. They are two types of multi-label learning algorithms: algorithm adaptation and problem transformation. Problem transformation approaches reduce a multi-label dataset to a single-label dataset or set of datasets. A single-label classifier is then run on each transformed dataset, and the results are aggregation-based. Algorithm adaptation, on the other hand, includes algorithms explicitly created to work with multi-label data [42].

In the statistics literature, active learning, a subfield of machine learning and artificial intelligence, is also known as “query learning” or “optimal experimental design”. Examples of data labels that can be applied for free or at a minimal cost are the “spam” flag placed on unwanted emails or the five-star rating given to films on social networking websites; the algorithm must choose the data from which it learns in order to perform better with less training. These flags and ratings are used by learning systems to filter the spam email better and suggest movies that the user might enjoy through the use of this information. Labelled instances are available for free in these cases, but in many more complex supervised learning tasks, they are challenging, time-consuming, or expensive to obtain. Labelling is a bottleneck in active learning systems. They use unlabelled instances as input to an oracle for labelling (e.g., a human annotator). With as few labelled examples as possible, the active learner can achieve high accuracy while reducing the cost of labelled data. For a large number of today’s machine learning problems, where data is plentiful but labels are scarce or expensive to obtain [5], active learning is an excellent solution. Figure 1 depicts a general framework for implementing active learning.

In traditional supervised learning, the model is trained using a pre-labelled training set that is not dynamic. Active learning (AL), on the other hand, is a branch of machine learning that allows classifiers to be built with fewer but more accurate data. AL focuses on applications where labelled data are scarce, but unlabelled data are easily accessible. It is challenging and frequently results in undesirable outcomes in constructing a predictive model using only the labelled data in these situations. Due to the potential financial and time commitment as well as the possibility that some data points are irrelevant to the model, obtaining all of the unlabelled dataset’s labels is not an option. This area of AL research aims to identify the most useful data points for labelling.

Active learning aims to improve a classifier by selecting unlabelled samples. Let us assume that ȹ is the AL process’s base classifier. Only a tiny number of pool-based AL scenarios $L_{s}$ have been labelled, although the number of unlabelled $U_{s\;}$ possibilities is enormous. On the other side, we have a technique that uses a selection criterion γ, such as an uncertainty measure, to choose a group of unlabelled cases. An AL procedure typically includes the following steps [30,42,43]:1γ choose examples from $U_{s}$ (unlabelled);2An annotator categorizes the chosen unlabelled instances;3Examples that were chosen are appended to $L_{s}$ then deleted from $U_{s}$;4ȹ is trained using the labelled set $L_{s}$;5The evaluation of performance for ȹ classifier is estimated;6Go to step 1, if no stopping condition.

In Figure 2, we describe how AL was employed in this paper. The first step was to build a predictive model using the labelled data. The label ranking classifier [44] is utilized in the proposed model. Data points are also marked to make them easier to understand. We used a criterion that measures the importance of the unlabelled dataset’s individual data points to make this choice. Many different criteria have been proposed, and many of them rely on the model’s inherent uncertainty. A popular baseline algorithm can select data points by uncertainty sampling near the model’s decision boundary. The proposed model uses a grid search for optimizing the label ranking classifier parameters. 

The parameters are the kernel, whose available values are {poly, RBF, linear}, degree = {1, 2, 3, 4}, and gamma, whose available values are {1, 3, 5, 7, 9}. The optimized parameters are set to {kernel = ’poly’, degree = 4, and gamma = 3}. These data points are then passed on to a domain expert for labelling. Predictive models are then built from scratch based on these newly labelled datasets. This process is carried out repeatedly until a predetermined endpoint is reached. Additionally, the criteria for determining when an application is complete may change. In some cases, new labels are purchased until the model performs well or the budget is exhausted.

One of the most critical aspects of an active learning algorithm is the design of appropriate criteria for selecting the most valuable instances for querying. Active learning algorithms commonly use informativeness and representativeness as two query selection criteria. An instance’s informativeness measures its ability to reduce a statistical model’s uncertainty and accurately represents unlabelled data input patterns [5].

## 3.3. Active Learning Selection Strategies 

There are many implemented strategies for active learning, such as active learning with instance selection, active learning by querying features, active learning for multi-label data, and different costs. We used the active learning algorithms for multi-label data according to the heart disease dataset. In this paper, active learning for multi-label data was implemented. The five selection methods (MMC, Random, Adaptive, QUIRE, and AUDI) applied are described in the following paragraphs. Table 1 shows a comparison of the implemented algorithms.

All strategies were tested on the heart disease dataset using 10-fold cross-validation. The AL experimental protocol iterative described in Algorithm 1 was applied each time a fold was carried out. The labelled set $L_{s}$ was constructed using a random selection of 5% $T_{r}\;($ the training set). Therefore, only a few labelled examples were used to train the initial classifier. The unlabelled set $U_{s}$ was derived from the $T_{r}$ examples that were not selected. The maximum iteration count β was set to 750. The multi-label classification algorithm was tested in each iteration by classifying $T_{s}$ (the test set). 

Table 2 shows the classification confusion matrix, including true positives, false positives, false negatives, and true negatives separately. As a result, a two-by-two confusion matrix (sometimes also referred to as a confusion matrix) was created [45]. By using the accuracy and the f1-measure, the classification performance could be studied in greater detail, as shown in Table 3.
**Algorithm 1.** AL experimental protocol.
***Inputs***: β→maximum number of iterationss→number of sampling examplesθ→oracle for labelling unlabelled examplesγ→multi-label AL strategyȹ→ multi-label classification algorithm&Ts→ a test set of multi-label examples&Tr→ a training set of multi-label examples**Begin**//Construct the labelled and unlabelled sets***L_s_***←**Resampl****e**⁡(***s*,*T_r_***);***U_s_***←***T_r_***\***L_S_***;//Train ȹ with ***L_S_***ȹ←**Trai n**(***L_s_***,ȹ);**for *i*** ter ←**1** to ***β*** do  //Select informative example from ***U_S_***  ***i***←SelectInformativeExample (***γ*,ȹ**,**U_S_**);  //label the selected example  Label(***θ***,***i***);  //Update the labelled and unlabelled sets  ***L_S_***←***L_S_***∪***i***;  ***U_S_***←***U_S_***\***i***;  //Train ← with ***L_S_***  ȹ←Train(***L_s_***,ȹ);  //Evaluate ȹ on ***T_S_***  Test (***T_s_***,ȹ);**end****end**

## 4. Experiment Discussion

### 4.1. Dataset Description and Experiment setup

The UCI machine learning repository was used to collect the heart disease dataset used in this study [23]. This repository, established in 1987, contains 487 datasets frequently consulted by students, educators, and researchers in machine learning. There are 303 instances of missing data in the Cleveland dataset, including 13 features, 1 target variable, and 20% of the total. The dataset consists of 138 normal instances versus 165 abnormal instances. The dataset contains approximately balanced instances by over 83.6% in the two categories of the class. Additionally, to overcome the minor rational imbalance, an election was performed to select balanced instances for validation purposes. Before beginning the data analysis, six missing instances were removed from this dataset. Details of the dataset can be found in Table 4.

The experiment was written and developed in Python 3.8 using an anaconda virtual environment with Intel Core i7 and 16GB RAM running on Microsoft Windows 10 x64-bit. Many dependencies have been used as Python modules, including SciPy, NumPy, pandas, Matplotlib, and ALiPy (AL in Python), which provide a module-based implementation of the AL framework, allowing for the easy evaluation, comparison, and analysis of the performance of AL approaches.

### 4.2. Dataset Analysis and Visualization Insights

This section provides a statistical description of the details of the heart disease dataset discussed in Table 5. Pair plots are an easy way to see how two variables are related. Each variable in the dataset is represented in a matrix of relationships that can be viewed instantly. It is also an excellent place to start when figuring out what kind of regression analysis to use. 

Figure 3 also shows the distribution of features in the heart disease dataset, which is particularly interesting. Figure 3 depicts the plotting of all of the features in the dataset (13 features). We can see that four attributes, namely age, trestbps, chol, and thalach, have a normal distribution. The most dominant value of gender is male, whereas in the CP attribute, the most frequent value is 0, and the lowest frequency is 3. Besides, Figure 3 illustrates that there are eight categorical attributes and six numeric attributes.

Figure 4 shows the heatmaps. These can be defined as visual representations of correlation matrices that show the relationship between multiple variables. In the range of −1 to 1, the correlation coefficient can take any value. If two variables are linearly linked, the statistical term for this relationship is a correlation. It can also be referred to as a correlation measure between two variables. In this scenario, the goal is to find a correlation between several variables and then organize the results. Here, a matrix data structure was used to store information. On a feature-by-feature basis, this is shown in Figure 4. The figure gives us many facts. Firstly, the five features showing the most class–feature dependence are {exang, cp, ddpeak, thalach, and ca} with correlations of 0.44, 0.43, 0.43, 0.42, and 0.39, respectively. The second fact denotes the feature–feature correlation shown in slope–ddpeak, thalach–age, slope–thalach, exang–cp, and exang–thalach with correlations of 0.58, 0.40, 0.39, 0.39, and 0.38, respectively. On the other hand, fbs, chol, trestbps, and restecg have the lowest correlation with the target.

### 4.3. Results

Since heart disease is an urgent medical topic, the experiment was designed to investigate the model’s accuracy and F-score to consolidate the recognition rate. The experiments included hyperparameter optimization using the grid search technique. Memorization still must to adapt to a similar new population rather than training. However, generalization can accept the challenge and act efficiently. 

Table 6 represents the average of five rounds of training pipelines running using the five different selection methods. Additionally, it compares the accuracy values of the selection methods before versus after the procedure of hyperparameter optimization using the grid search. We noted that the Adaptive selection method can achieve a higher accuracy rate for the learned model with a generalization concept than one with a memorization concept, which can be gained using classical ML model(s). Higher accuracy can even be obtained using the classical ML. 

Table 6 shows Adaptive is the minimal number of queries that results in the exact cost of MMC and is more significant than others by a single step with 51.4 ± 3% accuracy. The accuracy rate and generalization concept were enhanced due to the optimization of hyperparameters as 57.4 ± 4%. Figure 5 represents the complete profile of the Adaptive strategy method versus the others in terms of accuracy under the exact cost(s). The noticed insight was not stable for costs of less than 20. However, while the cost increased, the approximated insight became stable and valid. Figure 6 represents the complete profile overall experiment rounds of the Adaptive method compared to the others after hyperparameter optimization using the grid search regarding accuracy. It confirms the validity of the practical insight(s). However, memorization still represents a prominent peak of accuracy enhancement, consuming the exact same cost for every cost ≥40. Because recall is an important evaluation criterion of the classification model, the Adaptive method is capable of performing the number of queries that causes 62.2% and 65.2% of recall before and after the grid search optimization, respectively. 

Furthermore, Table 7 represents the average of five rounds of running the training pipeline using the different five selection methods in terms of F-score. Additionally, it compares the performance of the selection methods before versus after the hyperparameters’ optimization procedure by the grid search. We noted that the Adaptive selection method can achieve a higher F-score rate for the learned model with a generalization concept than one with a memorization concept for any classical ML model(s) before the grid search optimization. The Adaptive method reached 62.3 ± 4%, advancing ahead others by at least 1.26% of the generalization concept. However, as a response to the optimization process, the AUDI selection method achieved a better F-score. It reached 62.2 ± 3.6% of the F-score versus the Adaptive’s score, which decreased by 1.68%.

Figure 7 represents the complete profile of the Adaptive strategy method versus others in terms of F-score under the exact same cost(s). Unfortunately, the cost metric does not indicate a threshold of stability insight about the F-score of the Adaptive method. However, it performed better when the cost value was ≥ 60. During the cost increase, the approximated insight became stable and valid. Figure 8 represents the complete profile of the overall experiment rounds of the QUIRE method, comparing it with others after hyperparameter optimization using the F-score grid search. It confirms the validity of the practical insight(s) from Table 7, demonstrating a prominent, noticeable peak of F-score enhancement consuming the exact same cost for every cost ≥20. Overall, we can state that the accuracy was more regular and stable for the generalization concepts and increased due to hyperparameter optimization. To ensure the validity of the heart disease model, recall was recorded as 62.5% and 78.4% before and after optimization of the hyperparameters, respectively. Moreover, the average CPU time in seconds for running the MMC, Random, Adaptive, QUIRE, and AUDI methods was 0.73, 0.57, 20.65, 155.09, and 1.6, respectively.

### 4.4. Discussion 

Overall, the designed experiments included the impact tracking of the grid search method’s outcomes. The grid search optimized the hyperparameters of the different active learning selection methods. The fitness function of the optimization was subject to be indicated as accuracy or the F1-score. In addition, the recall was measured for the optimal selection method of active learning. In terms of accuracy optimization, the Adaptive method was the minimal number of queries that resulted in the exact cost of MMC and was more significant than others by a single step with 51.4 ± 3% accuracy. The accuracy rate and generalization concept were enhanced due to the optimization of hyperparameters as 57.4 ± 4% for every cost ≥40 and 62.2% and 65.2% of recall before and after the grid search optimization, respectively, with an average CPU time of 20.65 seconds. Additionally, in terms of F-score, the Adaptive method reached 62.3 ± 4%, advancing ahead of others by at least 1.26% and reaching 62.2 ± 3.6%, a decrease of 1.68%, with the generalization concept, respectively. Generally, the proposed learning cycle is considered to be an innovative contribution to medical heart disease diagnoses by using active learning approaches.

## 5. Conclusions

Heart disease is the most common cause of death in the world. Save a life by catching heart disease and related diseases such as dementia early on. To effectively treat patients before a heart attack, it is critical to predict heart disease accurately using a machine learning model. Active learning is a classification machine learning method that uses the generalization concept rather than the memorization concept available by regular classification algorithms. This paper utilized five multi-label active learning selection strategies, MMC, Random, Adaptive, Quire, and AUDI to query the most relevant data iteratively in order to reduce the cost of labelling. Additionally, the grid search methodology was applied to improve classification accuracy and the F-score in the instance of the lack of labelled data. The classification engine is based on a label ranking classifier used in each heart disease dataset strategy. According to the findings, the learning model could generalize beyond the sample data, with an accuracy and F-score of 57.4 ± 4% and 62.2 ± 3.6%, respectively. Moreover, the CPU elapsed time for the proposed model training is adequate. Moreover, there are open issues, including discretising the numeric values of features, categorization, and binning levels using advanced metaheuristic algorithms for fine-tuning the predictive models’ parameters and using enhancement classification algorithms rather than the label ranking classifier.

## Figures and Tables

**Figure 1 sensors-22-01184-f001:**
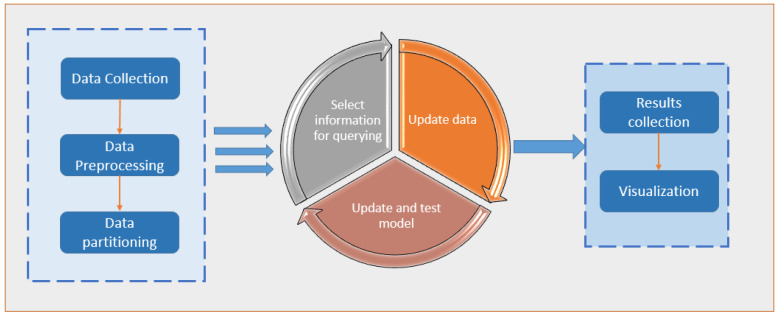
A general framework for implementing an active learning approach.

**Figure 2 sensors-22-01184-f002:**
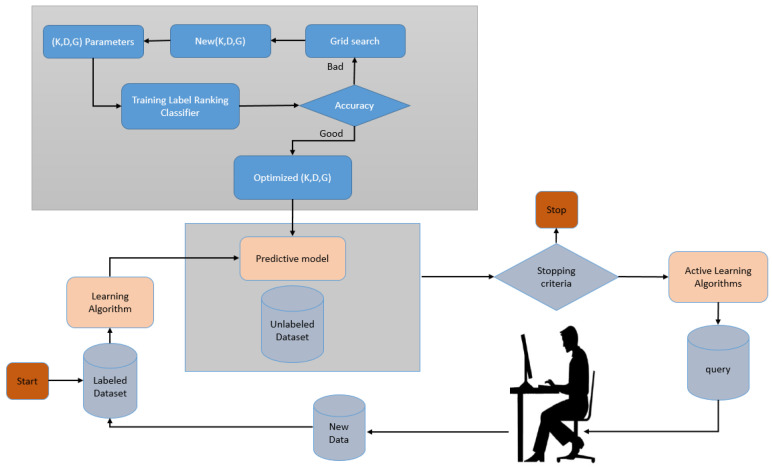
Implemented active learning cycle.

**Figure 3 sensors-22-01184-f003:**
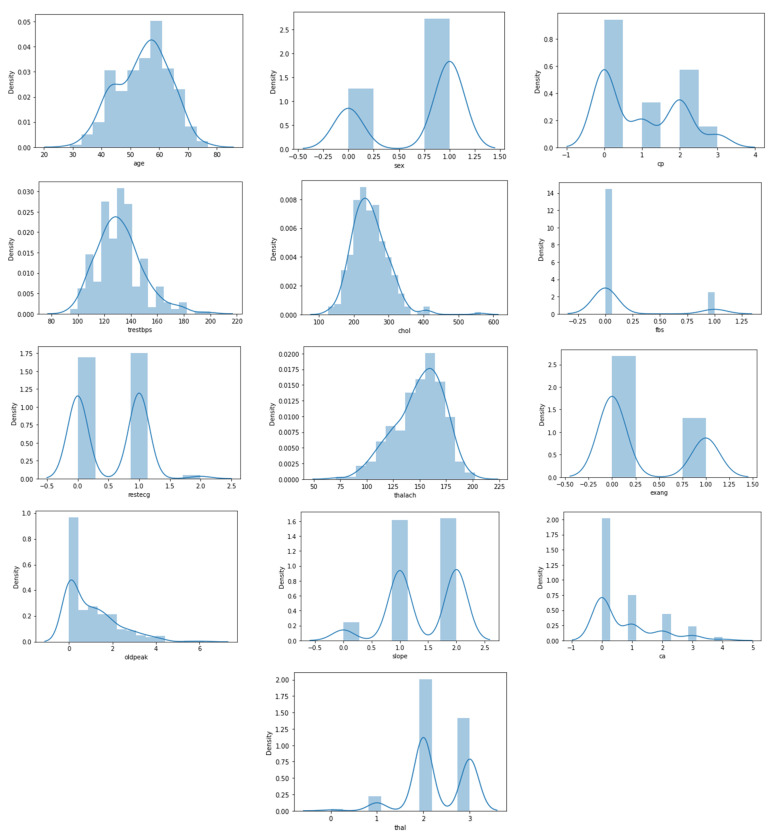
Statistical description of heart disease dataset.

**Figure 4 sensors-22-01184-f004:**
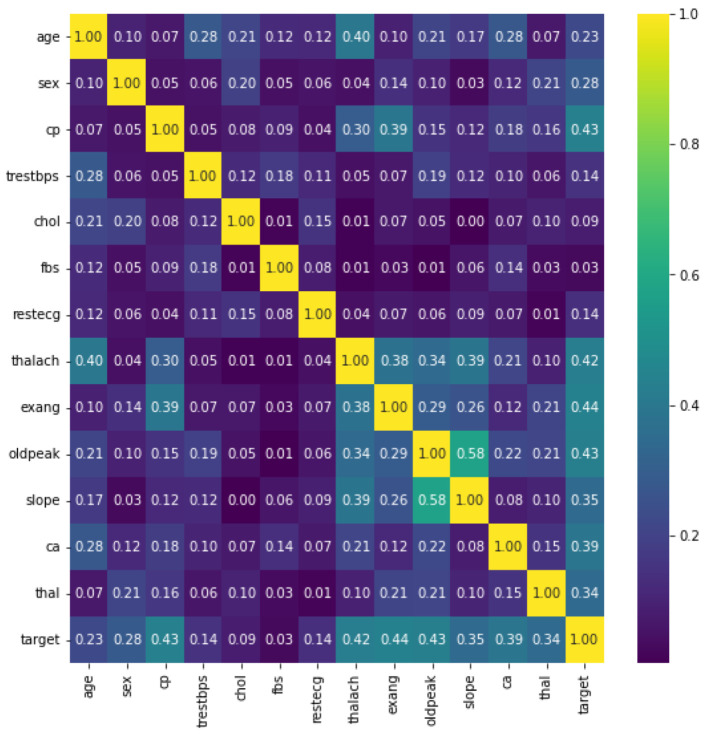
Dataset features heatmap.

**Figure 5 sensors-22-01184-f005:**
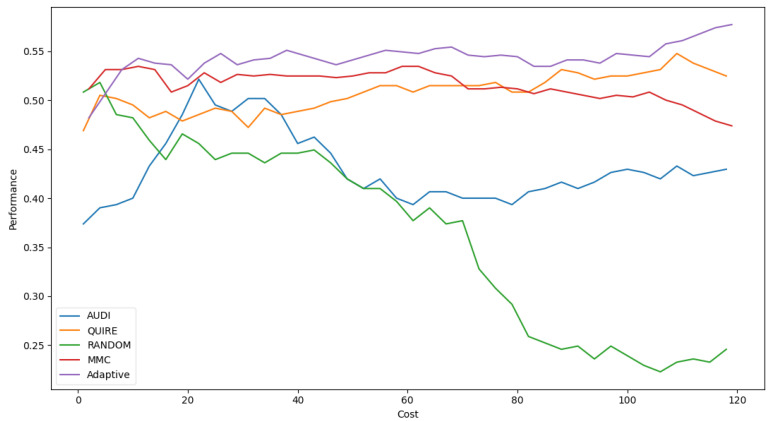
Classification accuracy for the five selection AL strategies without grid search optimizer.

**Figure 6 sensors-22-01184-f006:**
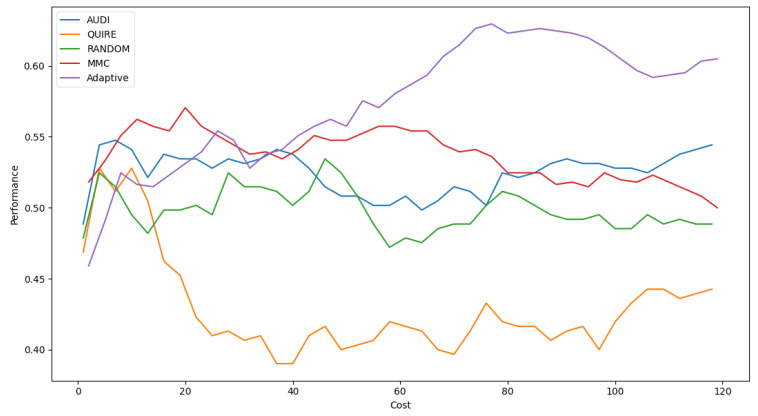
Classification accuracy for the five selection AL strategies with grid search optimizer.

**Figure 7 sensors-22-01184-f007:**
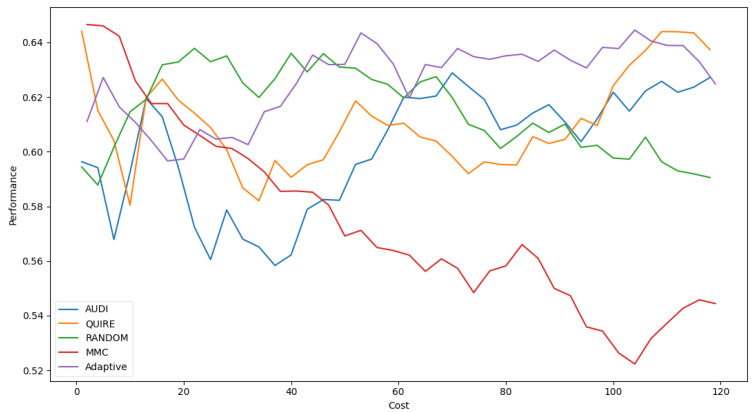
F1-measure for the five selection AL strategies without grid search optimizer.

**Figure 8 sensors-22-01184-f008:**
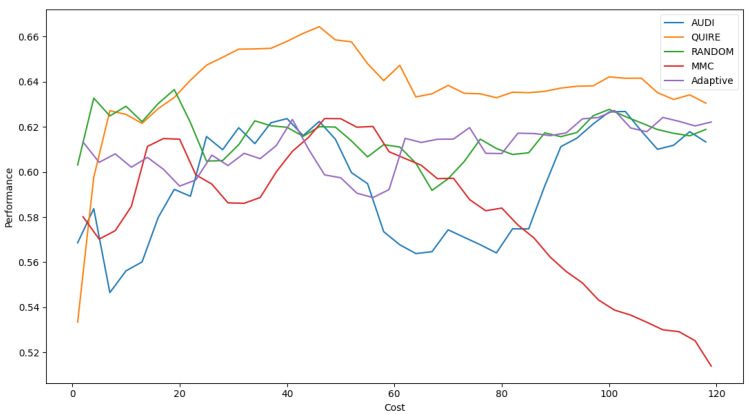
F1-measure for the five selection AL strategies without grid search optimizer.

**Table 1 sensors-22-01184-t001:** Multi-label query strategies.

Strategy	Description
**MMC**	Select an instance to run a loss reduction and confidence maximization query on all labels.
**Random**	Randomly select the instances or instance–label pairs.
**Adaptive**	The maximum margin uncertainty and label cardinality inconsistency are used to query all labels.
**QUIRE**	To choose a label–instance pair, consider the informational and representative qualities of the pair.
**AUDI**	Based on the degree of uncertainty and diversity, choose an instance–label pair.

**Table 2 sensors-22-01184-t002:** Confusion matrix.

Actual Output		Predicted Class	
		Positive	Negative
**Actual Class**	**Positive**	True Positives (TP)	False Negatives (FN)
	**Negative**	False Positives (FP)	True Negatives (TN)

**Table 3 sensors-22-01184-t003:** Performance evaluation metrics.

**Accuracy**	The ratio is defined as the correct outcomes of all the possible prediction values. Accuracy is the degree to which measures are within a specific range. At the same time, precision is the degree to which measurements are within.	Accuracy **$=\frac{TP+TN}{TP+TN+FP+FN}$**
**F1-score**	The weighted (sensitivity) and an accurate recall average are two different measures. F1 is a good option if you want to balance precision and reminder.	$F1\;\mathrm{Score}=\frac{\mathrm{TP}}{\mathrm{TP}+\frac{1}{2}\left(\mathrm{FP}+\mathrm{FN}\right)}$
**Recall**	The ratio correctly identified as diabetes in heart disease out of all heart disease instances.	$\mathrm{Recall}=\frac{\mathrm{TP}}{\mathrm{TP}+\mathrm{FN}}$

**Table 4 sensors-22-01184-t004:** Description of the dataset features.

Feature	Type	Description
**Age**	numeric	Years of age
**Sex**	categorical	1: male, 0: female
**CP**	numeric	Type of chest pain 1: typical angina 2: atypical angina 3: non-anginal pain 4: no symptoms
**Trestbps**	numeric	Standing blood pressure of the patient (in mm Hg)
**Chol**	numeric	Serum cholesterol (in mg/dl)
**Fbs**	categorical	If fasting blood sugar >120 mg/dL (1 = true; 0 = false)
**Restecg**	numeric	0: means “normal”. 1: averaging an aberrant ST-T wave (T wave inversions and/or ST elevation or depression of >0.05 mV) 2: demonstrating probable or definite left ventricular hypertrophy according to Estes’ criteria
**Thalach**	numeric	Attained maximum heart rate.
**Exang**	categorical	Angina induced by exercise (1 = yes; 0 = no)
**Oldpeak**	numeric	Exercise-induced ST depression in comparison to resting
**Slope**	numeric	The peak exercise’s slope ST-segment V 1: up-sloping 2: flat 3: down-sloping
**Ca**	numeric	Number of significant vessels (0–3) colored by fluoroscopy
**Thal**	numeric	3 = normal; 6 = fixed defect; 7 = reversable defect
**Num (target** **variable)**	categorical	Heart disease diagnosis (angiographic disease status) 0: less than 50% diameter narrowing 1: more than 50% diameter narrowing

**Table 5 sensors-22-01184-t005:** The feature distribution of the Cleveland Heart Disease dataset.

	Age	Sex	cp	trestbps	chol	fbs	restecg	thalach	Exang	oldpeak	slope	ca	thal	Target
**Mean**	54.3667	0.6832	0.9670	131.6238	246.2640	0.1485	0.5281	149.647	0.3267	1.0396	1.3993	0.7294	2.3135	0.5446
**Std**	9.08	0.47	1.03	17.54	51.83	0.36	0.53	22.901	0.47	1.16	0.62	1.02	0.61	0.50
**Min**	29	0	0	94	126	0	0	71	0	0	0	0	0	0
**Max**	77	1	3	200	564	1	2	202	1	6.2	2	4	3	1

**Table 6 sensors-22-01184-t006:** Comparative evaluation of active learning selection methods in terms of accuracy (the best values are bold).

Method	Before Hyperparameter Optimization			After hyperparameter Optimization		
	#Queries	Cost	Performance	#Queries	Cost	Performance
**AUDI**	121	121	0.431 ± 0.076	121	121	0.526 ± 0.048
**QUIRE**	121	121	0.508 ± 0.032	121	121	0.454 ± 0.068
**MMC**	73	122	0.476 ± 0.050	73	122	0.512 ± 0.050
**Adaptive**	**61**	**122**	**0.514 ±** **0.032**	**61**	**122**	**0.574 ± 0.020**
**Random**	121	121	0.355 ± 0.114	109	121	0.499 ± 0.068

**Table 7 sensors-22-01184-t007:** Comparative Evaluation of Active learning Selection Methods in terms of F-Score (the best values are bold).

Method	Before Hyperparameters Optimization			After Hyperparameters Optimization		
	#Queries	Cost	Performance	#Queries	Cost	Performance
**AUDI**	121	121	0.6014 ± 0.044	**121**	**121**	**0.6222 ± 0.036**
**QUIRE**	121	121	0.6104 ± 0.032	121	121	0.59158 ± 0.03
**MMC**	61	122	0.5734 ± 0.048	73	122	0.6070 ± 0.042
**Adaptive**	**62**	**122**	**0.6230 ± 0.040**	61	122	0.6062 ± 0.036
**Random**	121	121	0.6030 ± 0.028	109	121.2	0.6076 ± 0.036

## Data Availability

The Pima Indian Diabetes Dataset is available online for the research community at UCI Machine Learning Repository: Heart Disease Data Set (accessed on 1 December 2021).

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
