# Peer review of "Multi-Label Active Learning-Based Machine Learning Model for Heart Disease Prediction"

_sensors, 2022, doi:10.3390/s22031184_

Round 1

Reviewer 1 Report

The authors state that in this work, they aim to solve the problem of memorizing learning problem using active learning. However, their method and how this reduces the risk of overfitting are not clear in the paper. The introduction includes background details that are not relevant which has as a result, the focus of the paper is not clear i.e. lines 1-50.

The paper needs to be restructured to make more clear the goals. In the introduction focus on describing the problem of overfitting and how this can be reduced using active learning and how the five selection methods can help on the generalization of the learning. In another paragraph, you can give more details about the application domain which is the medical domain of cardiovascular diseases. 

Section 2: Describe related work. Use a different paragraph for describing the feature selection methods, a separate paragraph for active learning and optimization algorithms that are relevant to active learning, and one last paragraph for describing the application of Active Learning and ML in cardiovascular disease (or in health in general).

Section 3: Description of the specific methodology used in this work and on in general, the description of ML & Dataset Overview

Figure 4 is very small and it does not help in interpreting the results.

What is the difference between the results depicted in Figure 5 & Table 6?

Which are the features selected at the end using the 5 selection methods?

Author Response

Dear Reviewer 

We wish to thank you in advance for your time and effort. We wish to follow your comments, advice, completely reply to your questions and match your expectations. Please, kindly find the attached reputable response file for your comments. 

Reviewer 2 Report

This paper presents an application. Five existing methods (MMC, Random, Adaptive, QUIRE, and AUDI) in the literature were applied on a publicly-available dataset for multi-label active learning. The contribution of the paper is limited. For example, there is no new concept, new theoretical support, or new definition. 

Author Response

(The authors gave the same response as above.)

Reviewer 3 Report

For classification problems, accuracy and recall are important indicators. Especially for disease prediction in the medical field, accuracy and recall are the most important evaluation criteria of the classification model. F-score index is used in this paper, F-score is the index after the synthesis of accuracy and recall, which is more used to evaluate the comprehensive performance of the model. However, for heart disease prediction, any misclassification is intolerable (patients with disease are not detected, and patients without disease are judged as having disease), that is, the disease prediction model needs to ensure high accuracy and high recall. Therefore, is it reasonable for F-score to be used in the model evaluation of heart disease prediction task? Or is the classification model of heart disease detection task insufficient without using recall rate?

In the experiment of this paper, Cleveland dataset is selected for model training. Is the sample of Cleveland dataset balanced, that is, is the proportion of heart disease samples and non heart disease samples similar?

In practical problems, especially in disease prediction, the problem of sample imbalance is a common problem. Does the method proposed in this paper consider the problem of training sample imbalance?

In the introduction, the article lists some existing methods of heart disease prediction task. What are the characteristics of the proposed method compared with the existing methods? Is it superior in performance and shorter in training time?

What is the training time for each method? Can you compare the training time of the model?

Author Response

(The authors gave the same response as above.)

Round 2

Reviewer 1 Report

The authors address most of my comments, however, I still have some minor changes to recommend before recommending the manuscript for publication. As a first comment, the introduction still needs improvement in order to make clear the focus of the paper.

  • The first paragraph refers to personalized medicine but you are not actually applying your methods for personalized medicine, thus I will recommend removing this paragraph.
  • The second paragraph: "Active learning approaches..." - combine this paragraph with the 5th paragraph: "Active learning is one..". It is somehow a repetition.
  • "using artificial intelligence (AI), and ML approaches
    in healthcare throughout the last decade" - remove this line since you refer to a single reference that uses specifically logistic regression as a ML method.
  • "In this paper, the proposed model tries to solve the problem of memorizing learning
    models. Active learning is a perfect learning method if the model does not have emotions regarding data." - What do you mean "emotions regarding data"?
  • Section 2: Literature
    • Change it to "Related Work"
    • Table 1 is useful only if you are going to compare your results with these works at the end of the paper and if you are using ther same dataset. Otherwise, just refer to these works in text and specify what is the novelty and the benefits of your approach
    • Based on 2.2, the purpose of this paper is to use active learning applying to unlabelled data in combination with ML methods. Make it clear in the introduction
  • Section 3.1: ML definition and techniques are widely well known in this field and in the journal. Avoid to use this introduction but refer directly to the ML methods that you use in your approach
  • Dataset features: Have you tried to discretize the numeric values? It will be good to add it as future work, if not.
  • Page 11: Max line has been removed from table 6
  • Table 7 and general, include a comparison with the results applying to the same dataset without using active learning (maybe comparison with other works that use the same dataset)
  • "Unfourtunility" -> Unfortunately (page 16, under figure 9)
  • Section discussion is missing from this manuscript to justify the results and explain why they are better than with other methods. A comparison with other approaches is missing.

Author Response

(The authors gave the same response as above.)

Reviewer 2 Report

-

Author Response

(The authors gave the same response as above.)

Reviewer 3 Report

The "The empirical statistical analysis of heart disease using different visualization charts," should be "The empirical statistical analysis of heart disease using different visualization charts." So as to "Comparative evaluation of five active learning selection strategic methods verses the regular machine learning algorithms,".
"on algorithm. Error! Reference source not found. summ"??
"n in Error! Reference source not found., the go"??
"n. Error! Reference source not found. depic"??
"e found in Error! Reference source not found. and Error! Reference source not found.. "?

Author Response

(The authors gave the same response as above.)
